# Comparative analysis of novel hormonal agents in non-metastatic castration-resistant prostate cancer: A Taiwanese perspective

Po-Chieh Huang[1,☯], Li-Hua Huang[2☯], Cheng-Kuang Yang[1,3], Jian-Ri Li[1,3], Chuan-Shu Chen[1,4], Shian-Shiang Wang[1,4,5], Kun-Yuan Chiu[1,5], Yen-Chuan Ou[2]*, Chia-Yen Lin[1,4,6]*

1 Department of Urology, Taichung Veterans General Hospital, Taichung, Taiwan, 2 Division of Urology, Department of Surgery, Tungs' Taichung Metro Harbor Hospital, Taichung, Taiwan, 3 Department of Medicine and Nursing, Hungkuang University, Taichung, Taiwan, 4 School of Medicine, Chung Shan Medical University, Taichung, Taiwan, 5 Department of Applied Chemistry, National Chi Nan University, Nantou, Taiwan, 6 School of Medicine, National Yang Ming Chiao Tung University, Taipei, Taiwan

☯ These authors contributed equally to this work.
* lcyhank.tw@gmail.com (CYL); ycou228@gmail.com (YCO)

**Data Availability Statement:** All relevant data are within the manuscript and its Supporting Information files.

## Abstract

### Background

Non-metastatic castration-resistant prostate cancer (nmCRPC) is an asymptomatic condition with the potential to progress to metastasis. Novel hormonal agents (NHAs) are currently considered the gold standard treatment for nmCRPC, offering significant survival benefits. However, further evidence is needed to determine whether there are differences in the performance of these drugs among Asian populations.

### Methods

This retrospective analysis of nmCRPC patients aims to compare the efficacy and safety of three NHAs–apalutamide, darolutamide, and enzalutamide. Data were collected from two prominent prostate care centers in Taichung, Taiwan. Patient characteristics, treatment details, PSA responses, and adverse events were analyzed. Statistical comparisons were performed, and the study received Institutional Review Board approval.

### Results

Total of 64 patients were recruited in this study, including 29 darolutamide, 26 apalutamide, and 9 enzalutamide patients. Baseline characteristics varied between the three patient groups, but the treatment response still revealed similar results. The apalutamide group experienced more adverse events, notably skin rash. Discontinuation rates due to adverse events differed among the groups, and patients receiving darolutamide were less likely to discontinue treatment.

**Funding:** The author(s) received no specific funding for this work.

**Competing interests:** The authors have declared that no competing interests exist.

## Conclusion

This real-world study provides insights into NHA utilization in nmCRPC within the Taiwanese population. Adverse event profiles varied, emphasizing the need for individualized treatment decisions. The study underscores the importance of regional considerations and contributes valuable data for optimizing treatment outcomes in nmCRPC.

## Introduction

Prostate cancer stands as the most common solid cancer and the second most significant contributor to cancer-related mortalities in men across Western countries [1]. Despite undergoing surgical or radiotherapeutic treatments, approximately 15–30% of patients experience tumor recurrence [2]. Androgen deprivation therapy (ADT) to achieve castration status remains the primary method for treating recurrent diseases and situations where radical treatment is not suitable [3].

Patients diagnosed with non-metastatic castration-resistant prostate cancer (nmCRPC) are characterized by encountering increasing PSA levels despite maintaining testosterone at castration levels and lacking evidences of metastatic disease on conventional imaging modalities. Patients with a PSA doubling time (PSADT) of 10 months or less are considered to be at a heightened risk of rapidly developing metastases, which could potentially impact their survival. Thus, PSADT is regarded as the foremost predictor of disease progression and a key indicator for treatment [4]. However, most patients with nmCRPC do not exhibit significant clinical symptoms. Therefore, the treatment goals during this period aim not only to extend survival but also to preserve the patient's quality of life [5].

Novel hormonal agents (NHAs), including apalutamide, darolutamide, and enzalutamide, have emerged as the recommended treatment for patients with nmCRPC. These agents have not only demonstrated significant overall survival benefits, but also tolerable adverse effects [2]. Although all these NHAs predominantly target the androgen receptor signaling pathway and display significant efficacy against prostate cancer, it is important to note that they may manifest distinct profiles of adverse events [6]. Notably, the three pivotal trials exhibited substantial variations in reporting AEs and in the absolute risks of these events within the placebo arms. In the SPARTAN trial, apalutamide showed increased occurrences of fatigue, rash, weight loss, joint pain, falls, hypothyroidism, and fractures compared to the placebo [7]. Meanwhile, the PROSPER trial indicated higher rates of fatigue, hot flashes, hypertension, falls, major adverse cardiovascular events, and mental impairment disorders with enzalutamide compared to the placebo [8]. Within the SPARTAN and PROSPER trials, apalutamide and enzalutamide exhibited higher rates of fatigue, falls, and hypertension compared to the placebo [7]. On the other hand, enzalutamide and apalutamide can pass through the blood-brain barrier, whereas darolutamide has a reduced likelihood of crossing this barrier. Therefore, the ARAMIS trial reported only slightly elevated rates of fatigue, weakness, and rash, and did not show an increased occurrence of seizures, falls, fractures, cognitive disorders, or hypertension with darolutamide compared to the placebo [9, 10]. Interestingly, in Japanese patients, the occurrence of apalutamide-induced skin rash was more frequent (56%) compared to patients from other regions (23.8%) [11]. Despite a potential link between the incidence of skin rash and plasma exposure to apalutamide, it did not notably affect the severity of the rash [12]. Data indicates that ethnic variations and genetic diversity between Japanese and Caucasian populations might influence how patients with prostate cancer respond to treatments [13–15].

Presently, there are no direct comparisons available among these new NHAs, making it challenging to determine the most potent and tolerable drug among these three NHAs [2]. Additionally, there is an urgent need for more clinical research focusing on the efficacy and side effects of these drugs in Asian populations. This study aims to succinctly compare the effectiveness, safety, and AEs of the three NHAs. Furthermore, the investigation entails assessing the potential influence of diverse factors, including PSADT, PSA levels, and radiotherapy configuration, on these outcomes. Moreover, our objective is to elucidate particular clinical viewpoints regarding the use of these inhibitors in individuals diagnosed with nmCRPC, and to explore whether the outcomes in Asian patients align with those observed in Western populations.

## Materials and methods

The present study constitutes a retrospective analysis performed on a prospectively collected patient cohort from two distinguished prostate care medical centers situated in Taichung, Taiwan. The data extraction was conducted meticulously from well-maintained medical records. The protocol, bearing approval number CE21454B-1, received authorization from the Institutional Review Board of Taichung Veteran General Hospital. Since the analyses solely utilized deidentified data, the requirement for consent was waived by the ethics committee.

On October 15, 2023, we retrospectively accessed treatment data pertaining to the nmCRPC population from two hospital databases. We collected a series of 64 patients diagnosed with nmCRPC who received treatment involving three different NHAs from January 2015 to July 2023. Eligible participants encompassed adult patients with unequivocally confirmed nmCRPC diagnoses, embarking on ADT treatment protocols integrating apalutamide, darolutamide, or enzalutamide. Clinicopathologic data, comprising age, BMI, Gleason score, PSA levels, PSADT, initial clinical staging, duration of ADT treatment, and primary prostate cancer treatments, were comprehensively collected. Analytical data pertaining to PSA decline, adverse events associated with NHA treatment, and discontinuation of NHA were gathered for the three distinct NHA groups. This study solely focused on analyzing the initial drug used and did not assess sequential treatment. Furthermore, subsequent drug usage following a switch due to AEs was not included in this statistical analysis.

The administration of medications involved apalutamide dosed at 240 mg once daily, enzalutamide at 160 mg per day, and darolutamide at 600 mg twice daily among the three patient cohorts. Based on the regulations set forth by Taiwan's National Health Insurance Administration, the periods for which the three medications are covered by health insurance vary. Chronologically, the coverage initiation dates are as follows: Apalutamide began coverage on March 1, 2021, followed by darolutamide on November 1, 2021, and finally, Enzalutamide's coverage commenced on September 1, 2023. Notably, there exists a gap of over two years between the health insurance coverage initiation dates for Apalutamide and Enzalutamide.

Adverse events were recorded and graded in accordance with the National Cancer Institute criteria. The primary endpoints of this retrospective inquiry included the assessment of PSA response and the occurrence of adverse events. Initial cancer staging before the initiation of androgen deprivation therapy was conducted utilizing computed tomography scans and bone scintigraphy. In instances where patients lacked measurable lesions other than bone metastases, response classification was based on the absence of new lesions or disease progression. Among the 64 patients included in this study, 6 patients were identified as having metastasis. However, following ADT, evidence of metastasis could no longer be detected using conventional imaging modalities. According to the original definition of nmCRPC, these patients should have been excluded. However, in clinical practice, when we apply for medication

treatment, the lack of metastatic lesions means we can only follow the treatment options for nmCRPC.

Statistical analyses were executed using the Mann-Whitney U test for continuous variables and the Pearson's chi-squared test for categorical variables. A significance threshold of $P < 0.05$ was established to indicate statistical significance.

## Results

A total of 64 patients with nmCRPC received NHA treatment, with 29 receiving darolutamide, 26 administered apalutamide, and the remaining 9 given enzalutamide. Table 1 summarizes patient characteristics and primary clinical conditions before NHA initiation. Among the three groups, a notable difference in patient age was evident.

The darolutamide group comprised the oldest patients, averaging 79.69 years old. However, no significant age differences were observed between the enzalutamide and apalutamide groups. Assessing PSA levels at the commencement of NHA treatment revealed that the apalutamide group initiated therapy at an earlier stage of disease progression. Specifically, within the apalutamide group, 61.5% of individuals began treatment with PSA levels below 2,

**Table 1. Baseline characteristics of three novel hormonal agents in non-metastatic castration resistance prostate cancer (N = 64).**

|  | Daro | Apa | Enza | *p* value |
|---|---|---|---|---|
| N | 29 | 26 | 9 | |
| Age | 79.69±9.10 | 74.15±8.97 | 71.22±8.79 | 0.018* |
| BMI | 25.59±2.97 | 26.57±3.88 | 25.66±3.83 | 0.551 |
| Initial PSA before NHA index date | | | | 0.015* |
| <2 | 12(41.4%) | 16(61.5%) | 0(0.0%) | |
| 2~10 | 13(44.8%) | 7(26.9%) | 5(55.6%) | |
| >10 | 4(13.8%) | 3(11.5%) | 4(44.4%) | |
| Initial PSA before NHA index date (mean, SD) | 5.66±8.81 | 2.91±3.74 | 13.84±14.78 | 0.004** |
| Median PSA doubling time (median, IQR) | 4.5(3.5–8.5) | 4.0(2.7–6.1) | 2.8(2.8–2.8) | 0.214 |
| Initial clinical staging | | | | 0.002** |
| cT1-2N0M0 | 9(33.3%) | 9(56.3%) | 0(0.0%) | |
| cT3N0M0 | 11(40.7%) | 1(6.3%) | 1(12.5%) | |
| cT4N0M0 | 0(0.0%) | 0(0.0%) | 1(12.5%) | |
| cTanyN1M0 | 3(11.1%) | 6(37.5%) | 4(50.0%) | |
| cTanyNanyM1 | 4(14.8%) | 0(0.0%) | 2(25.0%) | |
| Gleason grade group at initial diagnosis | | | | 0.150 |
| 1~3 | 9(31.0%) | 4(16.7%) | 0(0.0%) | |
| 4~5 | 20(69.0%) | 20(83.3%) | 7(100.0%) | |
| Time from PC diagnosis to NHA index date, month (Mean, SD) | 80.17±64.78 | 76.95±53.17 | 38.23±26.66 | 0.128 |
| ADT duration, month, (Mean, SD) | 51.23±34.39 | 59.08±31.03 | 67.56±49.88 | 0.606 |
| Time from CRPC to index date, month (mean, SD) | 3.75±5.47 | 13.21±31.97 | 1.72±1.55 | 0.080 |
| Primary PC treatment | | | | |
| Radical prostatectomy | 12(41.4%) | 21(80.8%) | 6(66.7%) | 0.011* |
| Radiation therapy | 5(17.2%) | 7(26.9%) | 4(44.4%) | 0.247 |
| ADT alone | 9(31.0%) | 3(11.5%) | 1(11.1%) | 0.152 |
| Adjuvant or salvage radiation after radical prostatectomy(n, %) | 3(27.3%) | 10(47.6%) | 4(66.7%) | 0.273 |

Chi-square test or Kruskal-Wallis test.

*$p<0.05$

**$p<0.01$.

averaging 2.91, the lowest among the three groups. Clinical staging was earlier in the darolutamide and apalutamide cohorts, whereas 75% of the enzalutamide group showed evidence of lymph node involvement or distant metastasis at initial staging. A total of 19 patients had initially non-localized cancer, comprising 6 M1 and 13 N1 staging individuals, all meeting nmCRPC criteria before initiation of NHA therapy. Most patients had initial Gleason grade groups between four and five at diagnosis, indicating a higher-risk category. The duration from prostate cancer diagnosis to NHA treatment initiation was longer in the apalutamide and darolutamide groups, averaging five to six years, while the enzalutamide group had a comparatively shorter duration, averaging 31 months. This distinction arose from all patients in the enzalutamide group self-funding treatment before the coverage of the other two NHAs by health insurance. Additionally, the enzalutamide group initiated NHA treatment at a later clinical stage, specifically opting for enzalutamide. The ADT duration across the three groups showed no significant differences, averaging around fifty to fifty-five months. Radical prostatectomy remained the primary treatment choice across the groups. However, about 20% of patients received only ADT, predominantly observed among older individuals favoring darolutamide. The overall ratio for patients undergoing adjuvant salvage radiotherapy after prostatectomy was relatively low, standing at 43.6%.

Table 2 depicts the PSA response post-NHA treatment, alongside the number of AEs and treatment discontinuations.

Regardless of whether the criteria considered a 50% or a 90% decline in PSA at any point during the treatment period, no significant differences were observed among the three groups.

Fig 1 provides a detailed depiction of the PSA response observed in various NHAs at different time points, and illustrates the duration of effectiveness for those who continue medication. When considering a PSA reduction of 90% or more, patients continuing medication with apalutamide exhibit the quickest and highest proportion. After 6 months of treatment, 73.3% of patients who taking apalutamide achieve a PSA reduction of 90% or more, surpassing darolutamide's 60% and enzalutamide's 44.4%. However, when further followed up to 12 months, apalutamide and darolutamide showed similar performance. Regarding a PSA reduction of 50% or more, the majority of patients in all three groups experience this after 6 months of treatment, with enzalutamide having the highest proportion at 100%, followed by apalutamide at 93.3%.

Due to follow-up duration constraints, no significant difference existed in the time to discontinuation of NHA among the groups. However, in the Kaplan-Meier curves analysis of time to discontinuation of initial NHA (Fig 2), the darolutamide group significantly showed a better tolerability than the other two groups. The duration of drug exposure, in addition to

**Table 2. Treatment response of three novel hormonal agents in non-metastatic castration resistance prostate cancer N = 64.**

|  | Daro | Apa | Enza | *p* value |
|---|---|---|---|---|
| Treatment response of NHA |  |  |  |  |
| PSA90, % | 18(62.1%) | 18(69.2%) | 4(44.4%) | 0.415 |
| PSA50, % | 23(79.3%) | 22(84.6%) | 9(100.0%) | 0.328 |
| Time to discontinuation of initial NHA, month (median, IQR) | 13.8(7.6–16.3) | 7.7(2.9–24.0) | 18.7(12.1–27.3) | 0.179 |
| Adverse effect after NHA, any grade | 9(31.0%) | 18(69.2%) | 2(22.2%) | 0.006** |
| Drug discontinuation due to AE | 1(3.4%) | 12(46.2%) | 1(11.1%) | <0.001** |
| Follow up time | 15.5(9.0–23.1) | 27.3(24.0–41.5) | 24.7(16.3–40.7) | <0.001** |

Chi-square test or Kruskal-Wallis test.

*$p < 0.05$

**$p < 0.01$

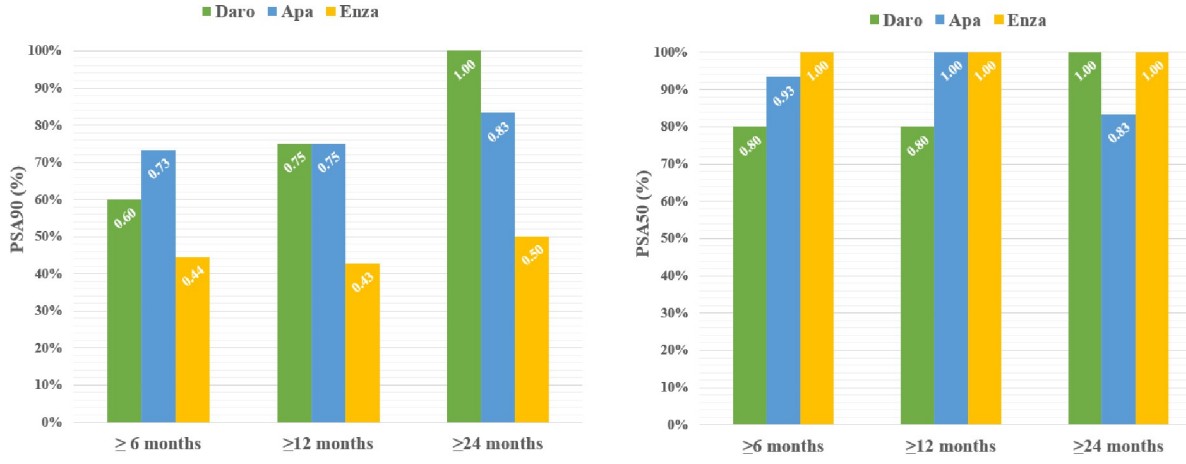

Supplement Table. InitialNHA_month

|  | ≥6M | ≥12M | ≥24M |
|---|---|---|---|
| PSA90, % | | | |
| Daro | 15 (60.0%) | 12 (75.0%) | 1 (100.0%) |
| Apa | 11 (73.3%) | 9 (75.0%) | 5 (83.3%) |
| Enza | 4 (44.4%) | 3 (42.9%) | 1 (50.0%) |
| PSA50, % | | | |
| Daro | 20 (80.0%) | 13 (81.3%) | 1 (100.0%) |
| Apa | 14 (93.3%) | 11 (91.7%) | 5 (83.3%) |
| Enza | 9 (100.0%) | 7 (100.0%) | 2 (100.0%) |

**Fig 1. The PSA response data of various NHAs at different time points.**

therapeutic efficacy such as PSA response, also includes drug-related adverse events, both of which may influence the decision to continue usage.

The incidence of AEs significantly differed across the groups, with the apalutamide group exhibiting the highest percentage at 69.2%. In particular, 12 patients from the apalutamide group stopped their treatment because of adverse events, making up 46.2% of that subset, and among them, 11 patients switched to darolutamide. In contrast, only one patient each from the darolutamide and enzalutamide groups ceased treatment due to AE. Commonly observed adverse events included dermatological manifestations such as skin rash, alongside physiological symptoms like fatigue, weakness, and decreased appetite, among other associated effects. Most AEs were categorized as grade 1 or grade 2 in severity. Notably, when considering more severe AEs classified as grade 3 and beyond, the apalutamide group exhibited the highest proportion, particularly concerning the occurrence of skin rash.

## Discussion

This study represents the first Taiwanese real-world exploration into the utilization of three prominent NHAs—apalutamide, darolutamide, and enzalutamide—in the treatment

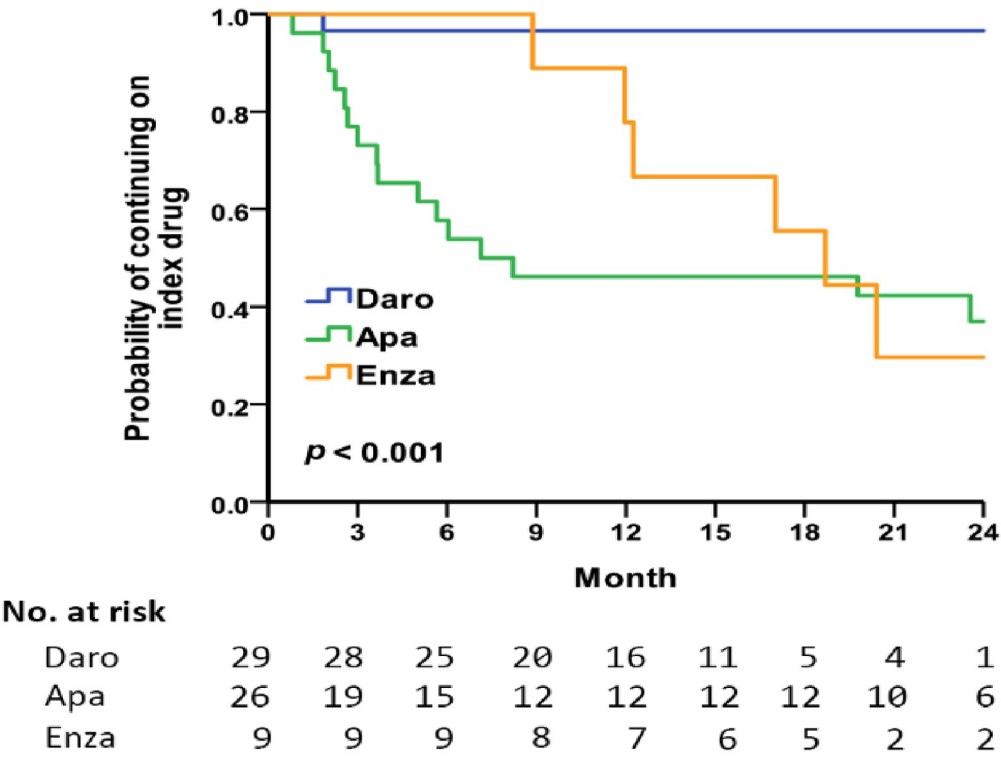

Overall survival

| | Total | Event | Censored | | Survival rate | | | | p for log rank |
|---|---|---|---|---|---|---|---|---|---|
| | | | n | % | 3m | 6m | 12m | 24m | |
| Drug | | | | | | | | | |
| Daro | 29 | 1 | 28 (96.6%) | | 96.6% | 96.6% | 96.6% | 96.6% | 0.001 |
| Apa | 26 | 16 | 10 (38.5%) | | 73.1% | 57.7% | 46.2% | 37.0% | |
| Enza | 9 | 7 | 2 (22.2%) | | 100.0% | 100.0% | 77.8% | 29.6% | |

**Pairwise Comparisons**

| | | Daro | Apa |
|---|---|---|---|
| Group | | Sig. | Sig. |
| Log Rank | Apa | .000 | |
| (Mantel-Cox) | Enza | .008 | .644 |

**Fig 2. Kaplan-Meier curves of time to discontinuation of initial novel hormonal agents in nmCRPC patients.**

landscape of nmCRPC. Apart from demonstrating promising therapeutic efficacy across all three drugs in Taiwanese patients, it has also revealed varying distributions of drug-related side effects, notably resembling observed patterns within Asian populations.

In a sequential series of pivotal studies, three phase 3 trials examined the efficacy of apalutamide, enzalutamide, and darolutamide in preventing metastatic occurrences among men with nmCRPC and PSADT ≤ 10 months [2]. Smith et al. reported from the SPARTAN trial that apalutamide significantly extended median metastasis-free survival (MFS) to 40.5 months compared to 16.2 months with placebo (HR 0.28, 95% CI 0.23–0.35, p < 0.001) [7]. Shortly after, Hussain et al. demonstrated from the PROSPER study that enzalutamide achieved a median MFS of 36.6 months versus 14.7 months in the placebo group (HR 0.29, 95% CI 0.24–0.35, p < 0.001) [8]. Subsequently, Fizazi et al. reported from the ARAMIS trial that darolutamide prolonged median MFS to 40.4 months compared to 18.4 months with placebo (HR 0.41, 95% CI 0.34–0.50, p < 0.001) at a median follow-up of 17.9 months [10]. These trials collectively underscore the notable efficacy of these NHAs in delaying metastatic progression among this patient population [2]. However, our study only captured an average discontinuation duration ranging from 7.7 to 18.7 months for these three medications. Given our relatively short follow-up period, further subsequent monitoring is essential to conduct a more comprehensive analysis of the efficacy outcomes.

Because such NHAs typically require prior application, it is common practice to administer ADT for approximately one month before commencing NHA treatment. Generally, at this stage, PSA levels have already begun to decline. This may consequently impact the response rate of PSA before and after NHA administration. In addition, analyzing the PSA90 characteristic, despite a significant number of cases in the apalutamide group in our cohort discontinuing treatment due to adverse effects, it is observed that among those who continue medication, a higher proportion achieves a PSA reduction of 90% compared to the darolutamide group in the early stages.

Darolutamide's limited ability to traverse the blood-brain barrier potentially contributes to a reduced occurrence of AEs observed across the three phase 3 trials [2, 9, 10]. Darolutamide, with its lower AE profile, is recommended for elderly patients, potentially accounting for the older age range in the darolutamide group. On the other hand, apalutamide and enzalutamide share structural similarities, possibly explaining the comparable discontinuation periods between the two groups in our study. When comparing the incidence of AEs among the three groups, the apalutamide group exhibited a higher rate of 69.2%. In the darolutamide group, only one patient ceased treatment due to adverse event of general weakness. No patients in the enzalutamide group discontinued treatment due to adverse events. Notably, skin rash constituted the highest proportion of grade 3 adverse events within the apalutamide group. Among apalutamide users, 12 discontinued treatment due to adverse events, primarily because of intolerable skin rash. Of these, 11 opted to switch to darolutamide, while another patient chose to discontinue treatment altogether. The extremely high discontinuation rate due to adverse events (AEs) among the apalutamide group might be attributed to the asymptomatic nature of nmCRPC. These patients demonstrate a comparatively lower tolerance towards side effects than those with metastatic hormone-sensitive prostate cancer (mHSPC). Particularly evident side effects, such as skin rash, may further prompt patients to consider switching medications or discontinuing treatment. However, both previous studies [11] and our results suggest a higher incidence of apalutamide-induced skin rash in Japanese and Taiwanese populations compared to Western groups, possibly indicating a higher susceptibility among Asian populations.

In our study, an examination of patient distribution among three distinct NHA groups revealed an unequal representation. Both the darolutamide and apalutamide groups included

at least twenty-five patients each, whereas the enzalutamide group comprised only nine patients. This discrepancy stems from the delayed inclusion of enzalutamide in Taiwan's health insurance coverage, regulated by the Bureau of National Health Insurance. Enzalutamide became eligible for coverage more than two years later than the other medications, resulting in fewer individuals choosing self-payment for its usage. Among our cohort of 28 patients, over one-third initiated NHA treatment when their PSA levels were below 2. This aligns with current clinical trends that no longer rely solely on a PSA greater than 2 as the defining criterion for CRPC. Initially, the 2008 Prostate Cancer Working Group 2 (PCWG2) defined CRPC based on a PSA level exceeding 2 ng/mL [16]. However, the 2015 PCWG3 consensus removed this specific PSA criterion [17]. This transition arose from recognizing the limitation of using solely a PSA level above 2 ng/mL as an indicator of disease progression, given its susceptibility to external influences [16]. PCWG3 emphasized a more comprehensive evaluation encompassing clinical symptoms, imaging findings, and overall clinical assessment beyond PSA levels [18]. This broader approach allows for a more holistic understanding of the disease, facilitating timely and precise treatment decisions. Early initiation of treatment often yields more favorable outcomes. Conversely, only 11 patients across the three groups began NHAs when their PSA levels were above 10, indicating a comparatively lower proportion opting for treatment at this higher PSA threshold. Based on the existing evidence, there is no significant difference among the three NHAs in the treatment selection for nmCRPC [2, 19]. The choice of medication currently relies on the preferences of physicians.

Regarding clinical staging, among the 64 patients, 19 initially presented with non-localized disease, indicating metastatic lesions that showed a complete response to ADT, as evidenced by the absence of visible tumors on conventional imaging tools. However, prostate-specific membrane antigen (PSMA)-PET demonstrates higher sensitivity and specificity in identifying metastases in contrast to conventional image, [20]. An earlier investigation revealed that PSMA-PET identified disease in 98% of nmCRPC cases and detected M1 disease in 55% of previously diagnosed nmCRPC patients, including subgroups with PSADT of $\leq 10$ months and Gleason score of $\geq 8$ [21]. Presently, PSMA-PET imaging exhibits superior diagnostic efficacy, yet its accessibility has been limited for most of our historical cases. The assessment of PSMA-PET imaging's utility in guiding treatments should be explored in forthcoming research endeavors. Although most patients did not undergo novel imaging like PSMA-PET imaging, the absence of diagnosing residual metastatic lesions via these means did not significantly impact medication selection. Hence, the decision against novel imaging for diagnosis was made at that point.

The majority of our patients exhibited a Gleason score between 4 and 5, indicating a high risk of local recurrence for most individuals. In primary treatment, most patients underwent radical prostatectomy. However, relatively fewer patients opted for adjuvant salvage radiation therapy post-surgery. The relatively low proportion of adjuvant or salvage radiotherapy may suggest a potential reduction in the subsequent development of nmCRPC among those receiving radiotherapy. However, such data might entail statistical selection bias. Despite salvage radiation therapy's potential to decrease nmCRPC incidence, its application poses risks of incontinence, frequency, and urgency, potentially leading to reduced patient willingness.

## Limitations

Our study is accompanied by several limitations that warrant consideration. The small sample size and differing baseline characteristics limited our ability to draw comprehensive conclusions. Moreover, the relatively short follow-up periods made it challenging to evaluate treatment efficacy, without considering survival impact. Additionally, our study only included

Taiwanese patients, so it may not be suitable to generalize our conclusions to the entire Asian population.

## Conclusion

In conclusion, our retrospective analysis investigated the usage of apalutamide, enzalutamide, and darolutamide in patients with nmCRPC, shedding light on treatment patterns, patient demographics, and adverse event profiles in the context of Taiwan's healthcare landscape. Notably, our findings emphasized the increased susceptibility of Asian populations to apalutamide-induced skin rash, aligning with existing literature. Despite these differences, no significant variations were observed in PSA response post-NHA treatment across the medication groups. While our findings corroborate previous research concerning apalutamide-induced skin rash, further investigations are warranted to delineate this phenomenon in Asian cohorts. Such insights can aid in refining therapeutic strategies and promoting better outcomes for patients receiving androgen-targeted therapies in this region.

## Supporting information

**S1 File. Raw data of nmCRPC in central Taiwan.**
(XLSX)

## Acknowledgments

I would like to extend my gratitude to the Biostatistics Group at the Department of Medical Research, Taichung Veterans General Hospital, for their valuable assistance in statistical analysis. Their expertise greatly contributed to the accuracy of our study. I appreciate their collaboration and support throughout this research.

## Author Contributions

**Conceptualization:** Jian-Ri Li, Shian-Shiang Wang, Yen-Chuan Ou, Chia-Yen Lin.

**Data curation:** Li-Hua Huang, Chuan-Shu Chen, Shian-Shiang Wang, Chia-Yen Lin.

**Formal analysis:** Chia-Yen Lin.

**Investigation:** Li-Hua Huang.

**Methodology:** Chuan-Shu Chen.

**Resources:** Li-Hua Huang, Cheng-Kuang Yang, Jian-Ri Li, Chuan-Shu Chen, Kun-Yuan Chiu, Chia-Yen Lin.

**Supervision:** Yen-Chuan Ou, Chia-Yen Lin.

**Validation:** Jian-Ri Li.

**Writing – original draft:** Po-Chieh Huang.

**Writing – review & editing:** Cheng-Kuang Yang, Yen-Chuan Ou, Chia-Yen Lin.

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
