## [Decision Letter · Decision Letter 0]

15 Apr 2024

PONE-D-24-05414Comparative Analysis of Novel Hormonal Agents in Non-Metastatic Castration-Resistant Prostate Cancer: A Taiwanese PerspectivePLOS ONE

Dear Dr. Lin,

Thank you for submitting your manuscript to PLOS ONE. After careful consideration, we feel that it has merit but does not fully meet PLOS ONE’s publication criteria as it currently stands. Therefore, we invite you to submit a revised version of the manuscript that addresses the points raised during the review process.

Overall, this study is well-conducted and informative. According to the reviewers' reports, a few comments need to be addressed through a more detailed discussion.

We look forward to receiving your revised manuscript.

Kind regards,

Wen-Wei Sung, M.D., Ph.D.

Academic Editor

PLOS ONE

Journal Requirements:

Reviewers' comments:

Reviewer's Responses to Questions

**Comments to the Author**

1. Is the manuscript technically sound, and do the data support the conclusions?

Reviewer #1: Yes

Reviewer #2: Yes

2. Has the statistical analysis been performed appropriately and rigorously? 

Reviewer #1: I Don't Know

Reviewer #2: Yes

3. Have the authors made all data underlying the findings in their manuscript fully available?

Reviewer #1: No

Reviewer #2: Yes

4. Is the manuscript presented in an intelligible fashion and written in standard English?

Reviewer #1: No

Reviewer #2: Yes

5. Review Comments to the Author

Reviewer #1: 1. Unmet Need Highlight:

o In the background section, the authors should address the unmet need more explicitly. Specifically, they could emphasize the lack of direct head-to-head comparisons among the three FDA-approved ARPis (darolutamide, apalutamide, and enzalutamide). Highlighting this gap would strengthen the rationale for their study.

2. Clarification on State Description:

o In line 21-22, the sentence mentions: “Non-metastatic castration-resistant prostate cancer (nmCRPC) refers to this state, which presents a potential threat to life.” However, the specific state is not clearly described. Please rephrase this sentence to provide clarity.

3. Support for Chemotherapy Use:

o Line 35-36 states: “Furthermore, when compared to chemotherapy, these treatments have demonstrated effectiveness in averting severe adverse effects [2, 7].” It’s essential to verify whether the cited references or guidelines indeed support the use of chemotherapy in nmCRPC status. I recommend asking the authors to reiterate this statement with appropriate evidence.

4. Replace “Review” with “Study”:

o In line 67, consider replacing “review” with “study” to avoid any confusion. This change will prevent readers from mistaking the article as a review article.

5. Clarification on nmCRPC Definition:

o Lines 104-107 discuss nmCRPC status. However, the usual definition excludes previously identified radiologically evident metastasis. The statements in this section seem to deviate from the standard definition. Please ask the authors to clarify this point for better alignment.

Reviewer #2: The manuscript examines the efficacy and safety of three novel hormonal agents (NHAs) – apalutamide, darolutamide, and enzalutamide. Data were gathered from two leading prostate care centers in Taichung, Taiwan. Patient characteristics, treatment specifics, PSA responses, and adverse events were meticulously analyzed. A total of 64 patients participated in this study, comprising 29 darolutamide, 26 apalutamide, and 9 enzalutamide patients. While baseline characteristics varied among the three patient cohorts, treatment responses yielded similar outcomes. The apalutamide group exhibited a higher incidence of adverse events, notably skin rash. Discontinuation rates due to adverse events varied among the groups, with patients receiving darolutamide displaying a lower likelihood of treatment discontinuation. These findings indicate the need to make individual therapy and optimize treatment plans for diverse patients.

Comments:

1. Think about demonstrating the PSA or other character changes across various treatments for greater clarity with figures.

2. Strengthen your conclusion by validating it with additional datasets.

3. Please carefully review and correct any typos and grammar errors.

6. PLOS authors have the option to publish the peer review history of their article (what does this mean?). If published, this will include your full peer review and any attached files.

Reviewer #1: **Yes: **Tzu-Ping Lin

Reviewer #2: No

---

## [Author Response · Author response to Decision Letter 0]

25 May 2024

Dear reviewer #1

We had revised our manuscript along with your suggestions as list below. We sincerely appreciate for your advices. 

1. Unmet Need Highlight:

In the background section, the authors should address the unmet need more explicitly. Specifically, they could emphasize the lack of direct head-to-head comparisons among the three FDA-approved ARPis (darolutamide, apalutamide, and enzalutamide). Highlighting this gap would strengthen the rationale for their study.

Answer: Thanks for your valuable comments. We have rewritten the background section, including the abstract and manuscript, to emphasize the rationale for this study.

2. Clarification on State Description:

o In line 21-22, the sentence mentions: “Non-metastatic castration-resistant prostate cancer (nmCRPC) refers to this state, which presents a potential threat to life.” However, the specific state is not clearly described. Please rephrase this sentence to provide clarity.

Answer: Thanks for your valuable comments. We have rewritten this section to address the definition of nmCRPC, the importance of PSADT, and the treatment goals for nmCRPC. (Page 5, line 17-25) 

3. Support for Chemotherapy Use:

o Line 35-36 states: “Furthermore, when compared to chemotherapy, these treatments have demonstrated effectiveness in averting severe adverse effects [2, 7].” It’s essential to verify whether the cited references or guidelines indeed support the use of chemotherapy in nmCRPC status. I recommend asking the authors to reiterate this statement with appropriate evidence.

Answer: Thanks for your valuable comments. We have removed the description about chemotherapy for nmCRPC. 

4. Replace “Review” with “Study”:

o In line 67, consider replacing “review” with “study” to avoid any confusion. This change will prevent readers from mistaking the article as a review article.

Answer: Thanks for your valuable comments. We have replaced all instances of "review" with "study" to avoid any confusion and ensure clarity.. 

5. Clarification on nmCRPC Definition:

o Lines 104-107 discuss nmCRPC status. However, the usual definition excludes previously identified radiologically evident metastasis. The statements in this section seem to deviate from the standard definition. Please ask the authors to clarify this point for better alignment.

Answer: Thanks for your valuable comments. We further clarified this clinical situation with the description: “According to the original definition of nmCRPC, these patients should have been excluded. However, in clinical practice, when we apply for medication treatment, the lack of metastatic lesions means we can only follow the treatment options for nmCRPC”. (Page 10, line 93-98)

 

Dear reviewer #2

We had revised our manuscript along with your suggestions as list below. We sincerely appreciate for your advices

1. Think about demonstrating the PSA or other character changes across various treatments for greater clarity with figures.

Answer: Thanks for your valuable comments. We have added a figure illustrating the PSA response data for various NHAs at different time points. Additionally, we have further clarified the PSA response on Page 14, lines 136-147.

2. Strengthen your conclusion by validating it with additional datasets..

Answer: Thanks for your valuable comments. We have revised the manuscript to include detailed PSA response data, thereby strengthening our conclusion.

3. Please carefully review and correct any typos and grammar errors.

Answer: Thanks for your valuable comments. We have carefully reviewed the manuscript and corrected all typos and grammatical errors using ChatGPT.

---

## [Editor Report · Decision Letter 1]

26 Jun 2024

Comparative Analysis of Novel Hormonal Agents in Non-Metastatic Castration-Resistant Prostate Cancer: A Taiwanese Perspective

PONE-D-24-05414R1

Dear Dr. Lin,

We’re pleased to inform you that your manuscript has been judged scientifically suitable for publication and will be formally accepted for publication once it meets all outstanding technical requirements.

Kind regards,

Wen-Wei Sung, M.D., Ph.D.

Academic Editor

PLOS ONE
---

## [Editor Report · Acceptance letter]

1 Jul 2024

PONE-D-24-05414R1 

PLOS ONE

Dear Dr. Lin, 

I'm pleased to inform you that your manuscript has been deemed suitable for publication in PLOS ONE. Congratulations! Your manuscript is now being handed over to our production team.

Kind regards, 

on behalf of

Dr. Wen-Wei Sung 

Academic Editor

PLOS ONE